# Evanescent Wave Sensitivity of Silica-Titania Rib Waveguides in the Single-Mode Propagation Regime

**Cuma Tyszkiewicz [1,*] and Paweł Kielan [2]**

[1] Department of Optoelectronics, Silesian University of Technology, Bolesława Krzywoustego 2 Street, 44-100 Gliwice, Poland

[2] Department of Mechatronics, Faculty of Electrical Engineering, Silesian University of Technology, 44-100 Gliwice, Poland; pawel.kielan@polsl.pl

* Correspondence: cuma.tyszkiewicz@polsl.pl

**Abstract:** The analysis reported in this paper shows that the homogeneous sensitivity of both fundamental rib waveguide modes, $HE_{00}$ and $EH_{00}$, can slightly exceed the sensitivity of the optimized parent slab waveguide. The most crucial difference in the behavior of these two polarizations is that the sensitivity of the $HE_{00}$ mode is the maximum for strip waveguides. In contrast, the sensitivity of the $EH_{00}$ mode can either decrease monotonically or not-monotonically with increasing rib height or behave like a homogeneous sensitivity characteristic of the slab waveguide's $EH_0$ mode. The second important conclusion comes from comparing the sensitivity characteristics with the distributions of the fundamental mode's optical power. Namely, the homogeneous sensitivity of the rib waveguide is at the maximum if, due to a slight variance in the cover refractive index, a variation in the weighted optical power carried by the mode is the maximum.

**Keywords:** optical design; rib waveguides; optical sensors; evanescent wave spectroscopy

## 1. Introduction

Rib waveguides play a fundamental role in planar integrated optical circuits (PICs) fabricated based on key material systems which include SOI (silicon on insulator) [1], InP/InGaAsP [2], AlGaAs-on-insulator [3], $Si_3N_4/SiO_2$ [4], a-$Al_2O_3/SiO_2$ [5], $SiO_2$-$TiO_2$/BK7 glass [6,7], and Ge-on-insulator [8]. This work concerns the silica-titania material platform. It is composed of $SiO_2$-$TiO_2$ waveguide films deposited on BK7 glass substrates. This platform has several advantages over mainstream material platforms and related technologies (i.e., SOI, $Si_3N_4$, and InP). They are discussed in the papers [9–11], where designs of integrated optics devices using the silica-titania material platform are proposed and discussed. Let us consider PICs used in constructing optical chemical and biochemical sensors [12] that rely on the principle of evanescent wave spectroscopy [13]. They include evanescent wave transducers. If those transducers integrate planar interferometers [14,15], specific requirements regarding the rib waveguides involved in their design can be formulated. In particular, rib waveguides should be in single mode to achieve high interferometric contrast [16] and have high homogeneous sensitivity. The latter quantity is a metric for measuring variations of guided mode effective refractive indices rendered by slight variations of the refractive index of a medium covering the waveguide [17]. The magnitude of homogeneous sensitivity depends on the magnitude and uniformity of the waveguide's refractive index and morphological parameters. It is well-established that the homogeneous sensitivity of slab waveguides, for a given polarization, has a single maximum if considered a function of the waveguide film thickness [17]. The paper [18] demonstrated that waveguides characterized by uniform refractive index profiles have higher homogeneous sensitivity than those with gradient-index refractive index profiles.

Moreover, homogeneous sensitivity increases with increased refractive index contrast on an interface between the waveguide and the cover. Assuming, therefore, that refractive indices of the rib waveguide and the substrate are determined by the given material system, the homogeneous sensitivity can be maximized by the careful selection of rib waveguide morphological parameters, namely: a parent slab thickness $H$, rib width $w$, and rib height $t$. Given this, one can see that for a given polarization, there is only one parent slab of optimum thickness $H_m$, which can serve as a reference for a whole family of rib waveguides described by $H$, $w$, and $t$. Their homogeneous sensitivity values can be normalized by dividing them by the sensitivity of the optimized parent slab.

The influence of morphological parameters of rib waveguides on their homogeneous sensitivity is discussed in papers [19,20]. However, the analysis presented in those papers concerns SOI, $SiO_xN_y$ (silicon oxynitride), and $Si_3N_4$ rib waveguides and does not consider the single-mode operation range. Moreover, it was carried in a limited range of morphological parameters, where the results obtained using the FEM (finite element method) and the EIM (effective index method) are compatible. These papers calculated homogeneous sensitivity from the waveguiding film's mode power flux and analytical equations derived from the EIM. The EIM only allows the correct derivation of modal field distributions if the mode is strongly confined to the rib [21]. In this respect, it is to be noted that such a mode is weakly sensitive to variations of the cover refractive index.

This article presents the results of the study on the relation between a rib waveguide single-mode operation regime and their homogeneous sensitivity in terms of characteristics of effective indices and homogeneous sensitivity as a function of their morphological parameters: $H$, $w$, and $t$. This work continues the analysis presented in our previous conference paper [22]. Nevertheless, we believe the results presented here are novel in two aspects. First, the research in this paper was carried out concerning the complete single-mode propagation condition described in the work [23]. Secondly, characteristics of homogeneous sensitivity are discussed concerning the distribution of waveguide mode optical power among a substrate, cover, and waveguide film. The literature has a well-established view that homogeneous sensitivity is the maximum if the fraction of optical power carried in the cover is maximized [19,24]. The presented results show that this is not the case considering rib waveguides. Based on the calculated characteristics of homogeneous sensitivity as a function of rib height, we have shown that homogeneous sensitivity is the maximum if the difference of weighted optical power carried by the mode is the maximum. The weights are refractive indices of the rib waveguide's cover, waveguide film, and substrate. The magnitude of the effective refractive index increases if the fraction of optical power carried in a medium having the highest refractive index increases. Typically, the waveguide film is such a medium. Given that the magnitude of the homogeneous sensitivity depends on the variation of the effective refractive index, our assumption is proper. That is because, due to the change in the cover refractive index, the distribution of the modal field changes, resulting in a modified distribution of optical power among the cover, waveguide film, and substrate, which have different refractive indices. A similar approach was adopted for the calculation of the rib waveguide's effective refractive indices by P.C. Kendall [25], who proposed weighting a value of the effective refractive indices by the magnitude of optical power in the rib waveguide's layers—a substrate, waveguide film, and cover.

## 2. Structure and Methods

Schematic diagrams of the structures being investigated in this work are presented in Figure 1. The slab waveguide comprises a silica-titania film deposited on a BK7 glass substrate. This film considered infinite along the axis $x$ is characterized by a thickness of $d$. Except for a strip area, the rib waveguide is fabricated from the parent slab through its partial or full thinning. As a result, a rib is formed. It is assumed that the rib's walls are vertical. A rib is characterized by width $w$ and height $t$. An analysis was carried out for the wavelength $\lambda = 635$ nm. However, in the case of the slab waveguides, the analysis was also

carried out for $\lambda$ = 1550 nm. This was carried out to compare the homogenous sensitivity of slab waveguides based on the SiO₂-TiO₂/BK7 material platform to those based on the SOI and Si₃N₄/SiO₂ platforms. The equations describing the refractive index chromatic dispersion of Si, SiO₂, and Si₃N₄ can be found in [26–28], respectively. The refractive indices of the silica-titania film $n_f$, BK7 glass substrate $n_s$, and cover $n_c$ for $\lambda$ = 635 nm are the same as those assumed in our paper on the single mode condition [19]. They are: $n_f$ = 1.8186, $n_s$ = 1.5150, and $n_c$ = 1.3320. Their values for $\lambda$ = 1550 nm are $n_f$ = 1.7595, $n_s$ = 1.5006, and $n_c$ = 1.3160.

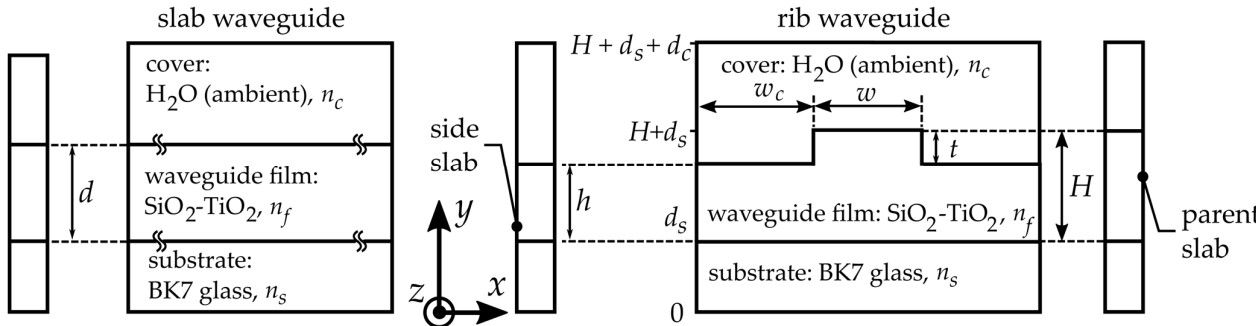

**Figure 1.** Schematic view of investigated slab and rib waveguides. Description of symbols: $d$—slab waveguide thickness, $d_c$ and $d_s$—substrate and cover layer thickness in a computational window, $w_s$—the distance between rib sidewalls and vertical boundaries of the computational window, $H$—parent slab thickness, $h$—side-slab thickness, $w$—rib width, $t$—rib height, $n_f$—refractive index of the waveguide film, $n_s$ and $n_c$—refractive indices of the substrate and cover, and $x$, $y$, $z$—coordinate system axes.

The characteristics of homogeneous sensitivity were derived based on the analysis of effective index (modal) characteristics of rib and slab waveguides. Homogeneous sensitivity is defined by the equation [17]:

$$S_H = \frac{dn_{eff}}{dn_c} \tag{1}$$

where $n_{eff}$ is an effective index of a given mode. The derivative (1) was approximated by a difference quotient, assuming that the increase of the cover refractive index is $\Delta n_c$ = 0.001. The effective index characteristics and optical power density, associated with fundamental rib waveguide modes, were calculated using the film mode matching method (FMM) [29] implemented in the FIMMWAVE 6.3 solver from Photon Design®. FMM is a rigorous and semi-analytical method that gives accurate results when applied to vertically walled rib waveguides. The Dirichlet boundary conditions were set on the boundaries of the computational domain, the size of which was selected so that the principal components of $HE_{pq}$ modes ($E_x$ and $H_y$) and $EH_{pq}$ modes ($E_y$ and $H_x$) were vanishing at those boundaries. The following values determining that size were adopted: $d_s$ = 3 µm, $d_c$ = 1 µm, and $w_c$ = 4 µm. The value of the parameter (1d)*nmode* was set to 60.

A schematic diagram presenting successive stages of the reported analysis is shown in Figure 2. Modal characteristics of slab and rib waveguides were calculated as a function of $d$ and $t$, respectively. A single mode propagation range for the investigated rib waveguides was determined based on the characteristics of two kinds of specific rib height values: $t_{sst}$ and $t_{cff}$. Namely, basic effective index characteristics of the $HE_{01}$ and $EH_{01}$ modes may, at $t_{sst}$, intersect the corresponding side-slab waveguide characteristics of the modes $HE_0$ and $EH_0$, respectively. If such an intersection occurs, the given first-order rib waveguide mode leaks to the side slab for $t < t_{sst}$, rendering the rib waveguide single mode in this interval. The second mechanism leading to single-mode propagation is to cut-off first-order modes. For $t > t_{cff}$, an effective index of the given first-order rib waveguide mode is less than the substrate refractive index. The procedure for determining the single-mode

propagation range based on the characteristics of $t_{sst}(w)$ and $t_{cff}(w)$ is described in detail in our previous paper [19]. Effective index characteristics of slab waveguides were calculated using the Effective Index Solver, also implemented in the FIMMWAVE solver.

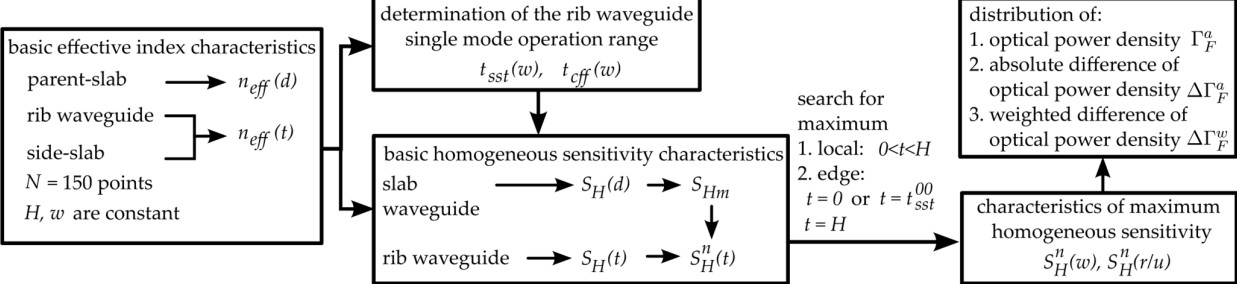

**Figure 2.** Schematic diagram of the analysis process. Description of symbols used in this figure: $n_{eff}$—effective refractive index, $t_{sst}$ and $t_{cff}$—characteristic values of the rib height, $S_H$ and $S_H^n$—absolute and normalized homogeneous sensitivity, $S_{Hm}$—homogeneous sensitivity of the optimized parent slab waveguide, and $r$ and $u$—effective rib width and height. A description of the morphological parameters is given in the caption of Figure 1.

The range of the rib waveguide's morphological parameter values in which the analysis was carried out is shown in Table 1.

**Table 1.** The range of morphological parameter values over which the analysis was carried out. Symbols $H$, $w$ and $t$ are explained in the caption to Figure 1.

| Parameter Name | Polarization | |
|---|---|---|
| | *HE* | *EH* |
| $H$ [nm] | 105, 120, 135, 140, 145, 150, 155, 160, 165, 170, 175, 180, 185, 195, 200, 210, 225, 240 | 135, 140, 145, 150, 155, 160, 165, 170, 170, 180, 185, 195, 200, 210, 225, 240, 270, 300, 330 |
| $w$ [μm] | From 1.0 to 2.0 with a step size of 0.02 | |
| $t$ [nm] | $N = 150$ equidistant values in the interval $t \in \langle 0, H \rangle$ | |

Each basic homogeneous sensitivity characteristic $S_H(t)$ was calculated from a pair of $n_{eff}(t)$ characteristics. The second characteristic in each pair was calculated for a value of the cover refractive index increased by a factor of $\Delta n_c = 0.001$. Those characteristics were grouped into sets according to a value of the parent slab thickness $H$ and searched for the presence of maximum values. Such a maximum can be located at the beginning of the interval <0,$H$>, ($t = 0$ − parent slab), at its end ($t = H$), and in the middle of it (local peak). Here, care must be taken if *EH* modes are being considered. This is because if $H$ is sufficiently low, the $EH_{00}$ mode can leak from the rib into the side slab (See Figure 1) for $t$ lower than some non-zero $t_{sst}^{00}$ value [23]. In that case, $t_{sst}^{00}$ is the left boundary of a search interval. The maximum sensitivity values are presented as sets of characteristics. They are functions of rib width $w$ or a parameter defined as the ratio between dimensionless side-slab thickness $r$ and dimensionless rib width $u$. The parameters $u$ and $r$, introduced in [30], are given by the following equations:

$$u = \frac{w + 2\gamma_c [k^2(n_f^2 - n_c^2)]^{-1/2}}{H + q} \tag{2}$$

$$r = \frac{h + q}{H + q} \tag{3}$$

$$q = \gamma_c [k^2 (n_f^2 - n_c^2)]^{-1/2} + \gamma_s [k^2 (n_f^2 - n_s^2)]^{-1/2} \tag{4}$$

where the parameter $\gamma$ depends on polarization, $\gamma_{c,s} = 1$ for $HE_{pq}$ modes, $\gamma_{c,s} = (n_{c,s}/n_i)^2$ for $EH_{pq}$ modes, and $k = 2\pi/\lambda$ where $\lambda$ is a wavelength. The sensitivity values were normalized to the sensitivity of the optimized parent slab because, for each polarization, there is only one value of slab waveguide thickness for which homogeneous sensitivity reaches its maximum.

Finally, the following parameters were determined to determine how the homogeneous sensitivity depends on a distribution of the waveguide mode optical power among the substrate, cover, and waveguide film: filling factor $\Gamma_F^a$, its difference $\Delta\Gamma_F^a$, and its weighted difference $\Delta\Gamma_F^w$. The first two parameters depend on the considered rib waveguide's region: the superscript $a$ accepts literals $c$ (cover), $f$ (waveguide film), and $s$ (substrate). The weighted difference $\Delta\Gamma_F^w$ measures the change of optical power density for the hole mode. The equation describes the filling factor:

$$\Gamma_F^A = \frac{\iint_A P_z(x,y) dx dy}{\iint_S P_z(x,y) dx dy} \tag{5}$$

where $A$ is the surface of a given part of the rib waveguide, $S$ is the surface of the computational window, and $P_z$ is the density of optical power defined by the Poynting vector, which has units of W/m².

$$P_z(x,y) = \left(\vec{E}(x,y) \times \vec{H}(x,y)\right) \circ \hat{z} \tag{6}$$

$\Delta\Gamma_F^a$ is also defined for a given region of the rib waveguide and is described by the equation:

$$\Delta\Gamma_F^a = \Gamma_F^{ai} - \Delta\Gamma_F^a \tag{7}$$

where $\Gamma_F^{ai}$ is the filling factor in the given region of the rib waveguide after increasing the cover refractive index from $n_c = 1.3320$ by $\Delta n_c = 0.001$. Finally, the weighted difference $\Delta\Gamma_F^w$ has the equation:

$$\Delta\Gamma_F^w = \frac{n_c \Delta\Gamma_F^c + n_f \Delta\Gamma_F^f + n_s \Delta\Gamma_F^s}{n_c + n_f + n_s} \tag{8}$$

## 3. Results

### 3.1. Parent Slab Waveguide

The characteristics of the homogeneous sensitivity for the parent slab as a function of the waveguide film thickness are presented in Figures 3 and 4. The ones presented in Figures 3b and 4 are for the wavelength $\lambda = 1550$ nm. It is well known that if a slab waveguide is single-layered, its characteristics have a single maximum for a specific thickness whose value depends on polarization ($d_{max}^{HE0}$ and $d_{max}^{EH0}$) [13]. Those values for the fundamental modes are specified in Table 2.

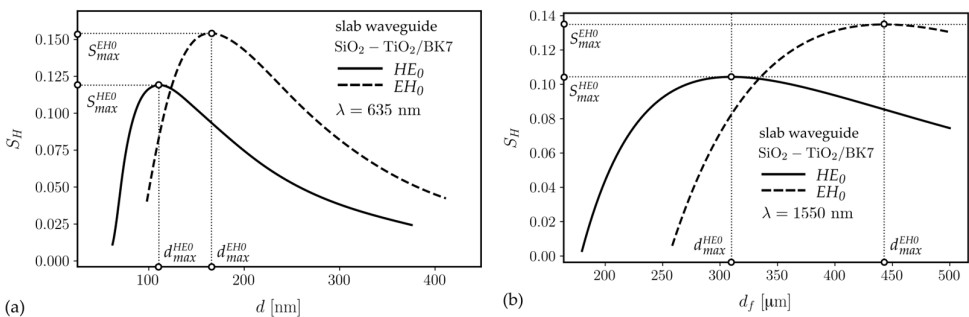

**Figure 3.** Homogeneous sensitivity characteristics of the SiO₂-TiO₂/BK7 slab waveguides for wavelengths $\lambda$ = 635 nm (**a**) and $\lambda$ = 1550 nm (**b**).

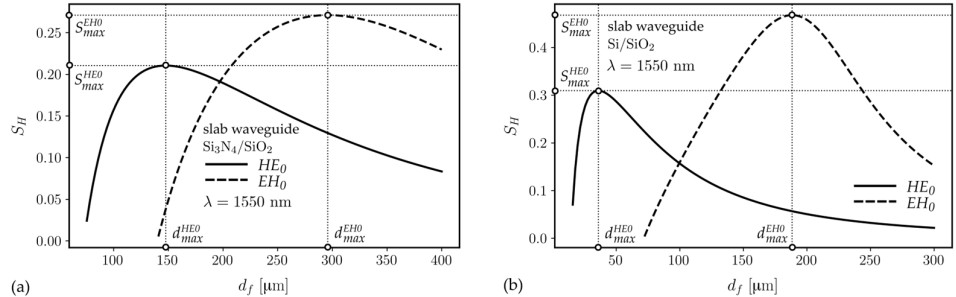

**Figure 4.** Homogeneous sensitivity characteristics of the Si₃N₄/SiO₂ (**a**) and SOI (**b**) slab waveguides for the wavelength $\lambda$ = 1550 nm.

**Table 2.** Optimized slab waveguide thickness $d_{max}$ and maximum homogeneous sensitivity $S_{max}$ of its fundamental modes for the silica-titania platform for the wavelengths 635 nm and 1550 nm and the Si₃N₄/SiO₂ and SOI platforms for the wavelength 1550 nm.

| Polarization | $HE_0$ | $EH_0$ |
|---|---|---|
| material platform: SiO₂-TiO₂/BK7, $\lambda$ = 635 nm | | |
| $d_{max}$ [nm] | 111 | 165 |
| $S_{max}$ | 0.119 | 0.154 |
| material platform: SiO₂-TiO₂/BK7, $\lambda$ = 1550 nm | | |
| $d_{max}$ [nm] | 310 | 443 |
| $S_{max}$ | 0.10 | 0.13 |
| material platform: Si₃N₄/SiO₂, $\lambda$ = 1550 nm | | |
| $d_{max}$ [nm] | 148 | 296 |
| $S_{max}$ | 0.21 | 0.27 |
| material platform: SOI, $\lambda$ = 1550 nm | | |
| $d_{max}$ [nm] | 36 | 188 |
| $S_{max}$ | 0.31 | 0.47 |

The sensitivity values of the fundamental HE and EH rib waveguide modes are normalized to the maximum sensitivity of the corresponding slab waveguide modes $S_{\max}^{HE0}$ and $S_{\max}^{EH0}$, respectively. The following equations give the normalized homogenous sensitivity of rib waveguides:

$$S_H^n = \begin{cases} S_H^{HE} / S_{\max}^{HE0} \\ S_H^{EH} / S_{\max}^{EH0} \end{cases} \tag{9}$$

As shown in Table 1, the fundamental slab modes became the most sensitive for significantly different values of the parent slab thickness. For this reason, the characteristics of the rib waveguide's homogeneous sensitivity were determined in two separate, although partly overlapping, ranges of the parent-slab thickness (see Table 1).

### 3.2. Rib Waveguide—Fundamental HE Modes

The selected representative characteristics of the $HE_{00}$ modes (quasi-TE) normalized homogeneous sensitivity $S_H^n(t)$ are presented in Figure 5. The range of single-mode propagation is distinguished by a bold line format. The character of changes in $S_H^n$ can be two-fold. The first can be observed on a set of characteristics for a value of the parent slab thickness smaller than $d_{max}$. They are presented in Figure 5a. One can see that $S_H^n(t)$ initially decreases, reaches the local minimum, and finally increases, but the maximum is not achieved for $t = H$. For each characteristic corresponding with the given value of $w$, the $S_H^n$ is maximum for $t = 0$. In the second case, the value of the parent slab thickness exceeds $d_{max}$, $S_H^n(t)$ increases with t, and reaches its maximum at $t = H$. The $HE_{00}$ mode of rib waveguides investigated in this paper acquires maximum homogeneous sensitivity for $H_o$ = 120 [nm], $w_o$ = 1.0 [μm], and $t_o$ = 120 [nm].

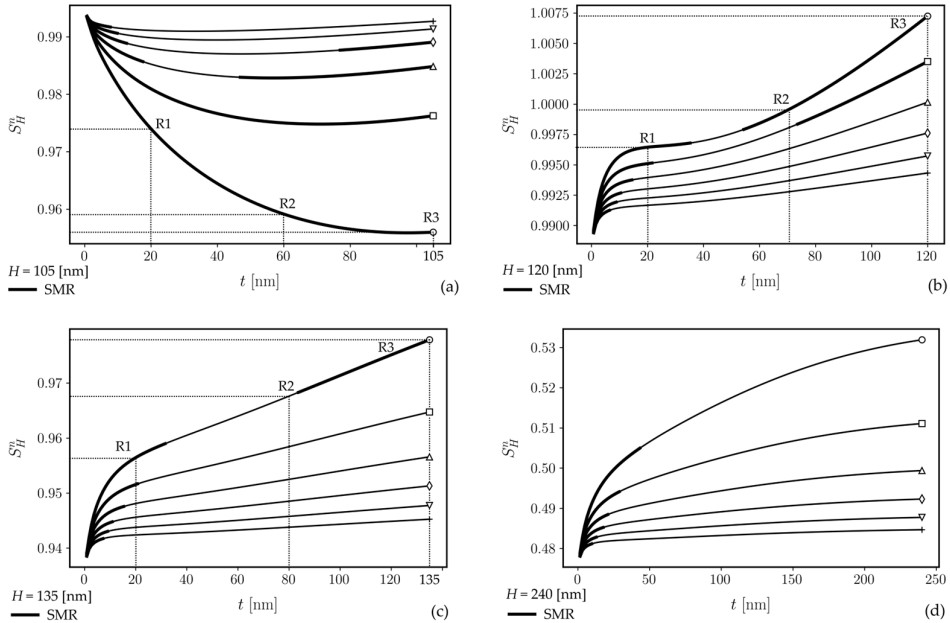

**Figure 5.** Normalized homogeneous sensitivity characteristics of $HE_{00}$ modes for four selected values of the parent slab thickness $H$: 105 nm (**a**), 120 nm (**b**), 135 nm (**c**), and 240 nm (**d**). Marker symbols: ∘–$w$ = 1.0 [μm], □–$w$ = 1.2 [μm], △–$w$ = 1.4 [μm], ◊–$w$ = 1.6 [μm], ▽–$w$ = 1.8 [μm], and +–$w$ = 2.0 [μm]. The bold line format indicates the single-mode propagation regime (SMR).

The characteristics of homogeneous sensitivity concerning the rib width w for a maximum value of the rib height are presented in Figure 6. Here, the bold line format implies that $S_H^n$ is the maximum. One can observe that sensitivity reaches the maximum value almost exclusively if $t = H$. All characteristics in Figure 6 are plotted using the bold line format except for the curve for $H$ = 105 nm. It is also apparent that there are no local maxima of $S_H^n$ and except for $H$ = 105 nm, which is smaller than optimized thickness $d_{max}$ for the slab waveguide's $HE_0$ mode, and that $S_H$ is decreasing with the increasing rib width $w$.

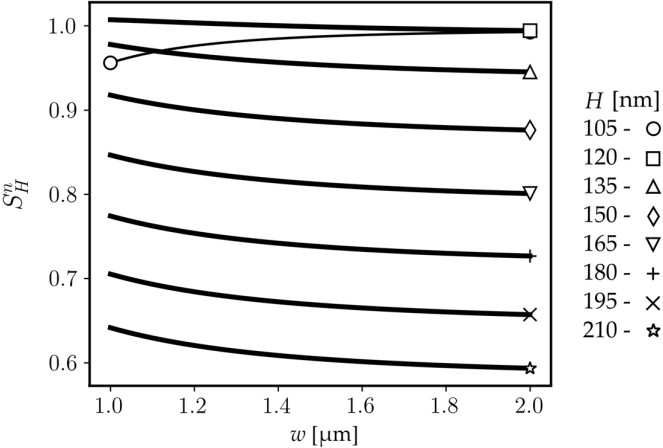

**Figure 6.** Normalized homogeneous sensitivity of $HE_{00}$ modes for $t = H$. The bold line format is used if the sensitivity reaches the maximum value.

Discussing the rib waveguide sensitivity characteristics concerning their morphological parameters is interesting, given how the optical power is distributed among the substrate, cover, and waveguide film. Three values of the parent slab thickness H were selected, and for each of them, the three values of *t* were chosen from the appropriate $S_H^n(t)$ characteristic. They are designated in Figure 5a–c by symbols R1, R2, and R3, respectively. Each corresponds to a rib waveguide with a tuple (*H*, *w*, *t*). The selected parameters are listed in Table 3.

**Table 3.** Morphological parameters of the rib waveguides selected for the determination of $HE_{00}$ modes $\Gamma_F^a$, $\Delta\Gamma_F^a$, and $\Delta\Gamma_F^w$.

| H [nm] | w [μm] | t [nm] | | |
|--------|--------|--------|--------|--------|
| 105 | 1.0 | R1: 20.0 | R2: 60.0 | R3: 105.0 |
| 120 | 1.0 | R1: 20.0 | R2: 70.0 | R3: 120.0 |
| 135 | 1.0 | R1: 20.0 | R2: 80.0 | R3: 135.0 |

The characteristics of the calculated values of the filling factor and its difference are presented in Figures 7a and 7b, respectively. One can observe that $\Gamma_F^c$ is not monotonically increasing with the homogeneous sensitivity. In all cases, the optical power is transferred from the substrate to the waveguide film and cover layer— $\Delta\Gamma_F^s < 0$, $\Delta\Gamma_F^f > 0$, and $\Delta\Gamma_F^c > 0$.

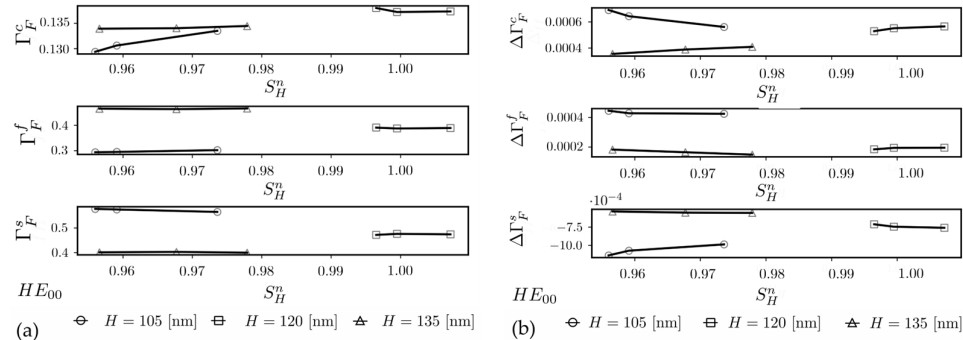

**Figure 7.** Characteristic of *HE*₀₀ modes' filling factor $\Gamma_F^a$ (**a**) and its absolute difference $\Delta\Gamma_F^a$ (**b**), both as a function of the normalized homogeneous sensitivity. The superscript a accepts literals c (cover), f (waveguide film), and s (substrate).

The characteristic of the weighted absolute difference of the filling factor is presented in Figure 8. It demonstrates two things. First, they $\Delta\Gamma_F^w$ can be either positive or negative. The positive value is obtained for the lowest parent slab thickness H value. This means that the optical power is predominately transferred from the substrate to the waveguide film due to the increase in the cover refractive index and a transformation of the *HE*₀₀ modal field. For higher values of H, the $\Delta\Gamma_F^w$ is negative. It indicates that the optical power more intensively flows to the cover, which has the lowest refractive index. The other conclusion derived from the characteristic of $\Delta\Gamma_F^w$ is that its absolute value monotonously increases with the normalized homogeneous sensitivity.

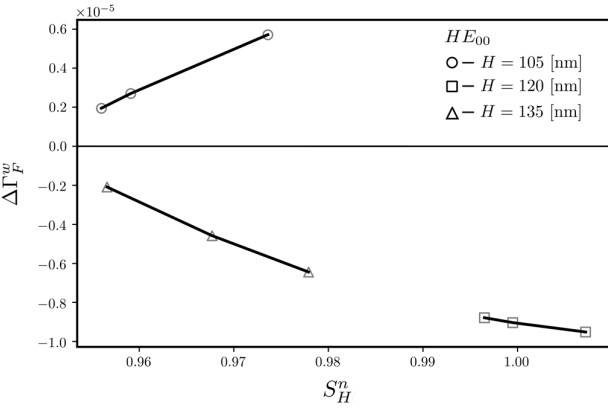

**Figure 8.** Characteristic of *HE*₀₀ modes' filling factor weighted difference $\Delta\Gamma_F^w$ as a function of the normalized homogeneous sensitivity.

### 3.3. Rib Waveguide—Fundamental HE Modes

The selected representative characteristics of *EH*₀₀ modes normalized homogeneous sensitivity $S_H^n(t)$ are presented in Figure 9. The range of single-mode propagation is distinguished by a bold line format. One can observe that characteristics behave differently than those of *HE*₀₀ modes. As one can observe from Figure 8c,d, they have a single maximum, for some *t* inside its range of variability, if the parent slab thickness exceeds the optimized thickness *d*ₘₐₓ for the slab waveguide's *EH*₀ mode.

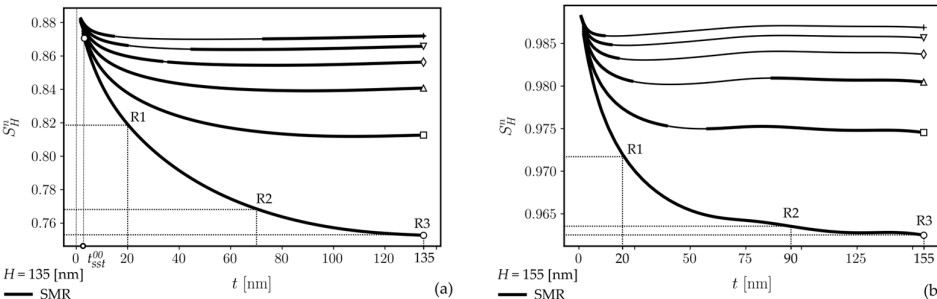

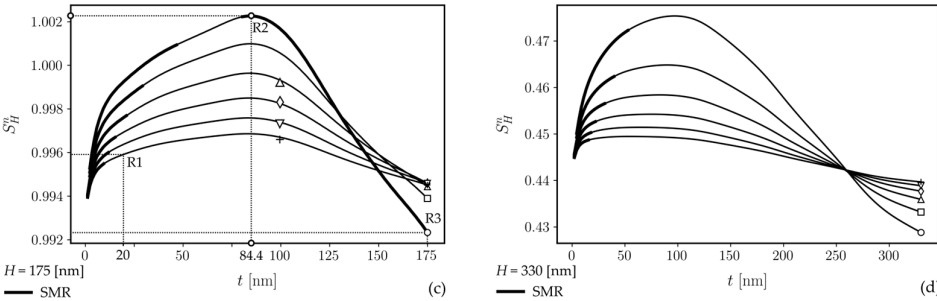

**Figure 9.** Normalized homogeneous sensitivity characteristics of $EH_{00}$ modes for four selected values of the parent slab thickness $H$: 135 nm (**a**), 155 nm (**b**), 175 nm (**c**), and 330 nm (**d**). Marker symbols: $\circ$–$w$ = 1.0 [μm], $\square$–$w$ = 1.2 [μm], $\triangle$–$w$ = 1.4 [μm], $\lozenge$–$w$ = 1.6 [μm], $\triangledown$–$w$ = 1.8 [μm], and +–$w$ = 2.0 [μm]. The bold line format indicates a single-mode propagation regime (SMR).

The homogeneous sensitivity characteristics of $EH_{00}$ modes for $t = H$ are presented in Figure 10a. They behave similarly to those obtained for $HE_{00}$ modes (Figure 6). The only difference is that the rate of $S_H^n(w)$ decrease is smaller. One can observe that the maximum sensitivity of $EH_{00}$ modes is predominately reached inside the range of variability of t, i.e., there is a single local maximum. Local maxima of $S_H^n(t)$ exist only for $EH_{00}$ modes. Their characteristics, presented in Figure 10b, concern the dimensionless parameter $r/u$, which reflects the transition from a strip waveguide (low r/u) to the structure tending to the slab waveguide (high $r/u$). For the given parent slab, the maxima of $S_H^n$ increase as the rib waveguide structure tends to the slab. The only exception is the characteristic for $H = 175$ nm on which the global maximum of $S_H^n$ is obtained. For this value, the maximum of $S_H^n$ is independent of r/u. The bold line format in Figure 9b implies that $S_H^n$ is the maximum in the whole range of $t$ variability.

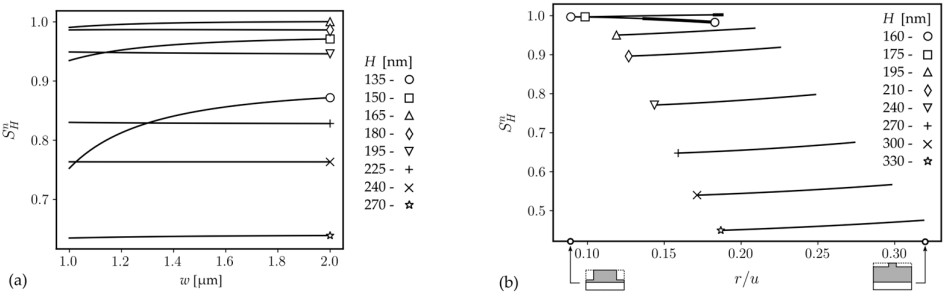

**Figure 10.** Normalized homogeneous sensitivity of $EH_{00}$ modes for $t = H$ (**a**) and its local maximum concerning dimensionless parameter $r/u$ (**b**).

The values of $\Gamma_F^a$, $\Delta\Gamma_F^a$, and $\Delta\Gamma_F^w$ have been calculated by the same process as the one used for the $HE_{00}$ modes. The selected morphological parameters are listed in Table 4.

**Table 4.** Morphological parameters of the rib waveguides selected for the determination of $EH_{00}$ modes $\Gamma_F^a$, $\Delta\Gamma_F^a$, and $\Delta\Gamma_F^w$.

| $H$ [nm] | $w$ [μm] | $t$ [nm] | | |
|---|---|---|---|---|
| 135 | 1.0 | R1: 20.0 | R2: 70.0 | R3: 135.0 |
| 155 | 1.0 | R1: 20.0 | R2: 90.0 | R3: 155.0 |
| 175 | 1.0 | R1: 20.0 | R2: 84.4 | R3: 175.0 |

The characteristics of the calculated values of the filling factor and its difference are presented in Figure 11a,b, respectively. As one can see, similarly as is the case for $HE_{00}$

modes, $\Gamma_F^c$ is not monotonically increasing with the homogeneous sensitivity, and in all cases, the optical power is transferred from the substrate to the waveguide film and cover layer—$\Delta\Gamma_F^s < 0$, $\Delta\Gamma_F^f > 0$, and $\Delta\Gamma_F^c > 0$.

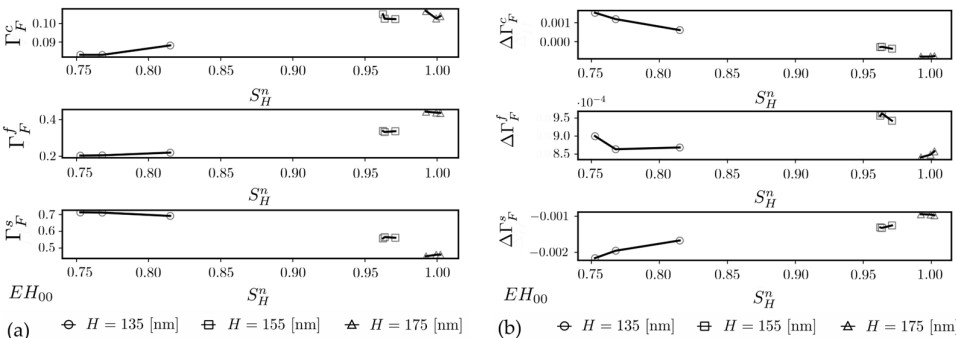

**Figure 11.** Characteristics of $EH_{00}$ modes' filling factor $\Gamma_F^a$ (**a**) and its absolute difference $\Delta\Gamma_F^a$ (**b**), both as a function of the normalized homogeneous sensitivity. The superscript a accepts literals c (cover), f (waveguide film), and s (substrate).

The characteristic of the weighted absolute difference of the filling factor is presented in Figure 12. Contrary to the situation observed for $HE_{00}$ modes (Figure 7), the $\Delta\Gamma_F^w$ is positive for all considered values of the parent slab thickness. As one can observe, $\Delta\Gamma_F^w$ monotonously increases with the homogeneous sensitivity.

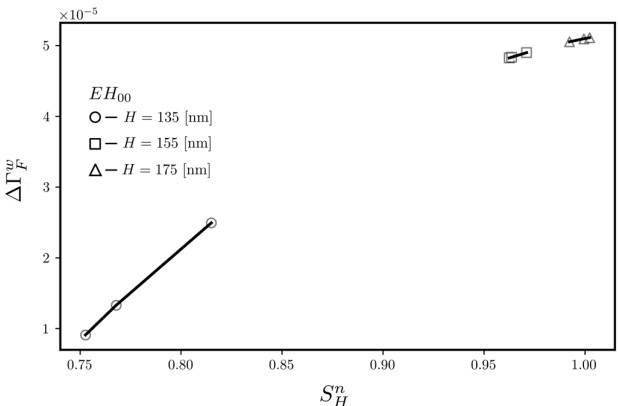

**Figure 12.** Characteristic of the weighted difference $\Delta\Gamma_F^w$ as a function of the normalized homogeneous sensitivity of $EH_{00}$ modes.

Regardless of the fabrication technology of rib waveguides, their final opto-geometrical parameters are more or less different than originally assumed. In order to assess how strongly variations of opto-geometrical parameters will influence the homogeneous sensitivity, the following procedure was adapted. One rib waveguide was selected for each fundamental mode. Their parameters are listed in Table 5. For theme, values of base homogeneous sensitivities, $S_H^{(1)}$ were calculated. Subsequently these parameters were individually increased by $p$ percent. This way, the second value $S_H^{(2)}$ was obtained for rib waveguides having one parameter increased. The relative change $\delta S_H$ was calculated using the equation below:

$$\delta S_H = 100\frac{\left|S_H^{(2)} - S_H^{(1)}\right|}{S_H^{(1)}}[\%] \tag{10}$$

The characteristic $\delta S_H(p)$ are presented in Figure 13.

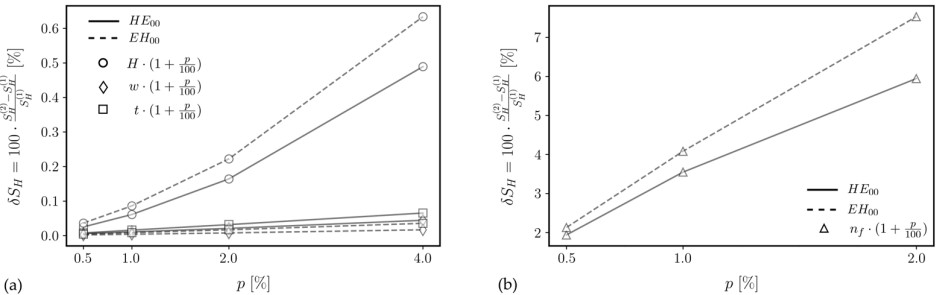

**Figure 13.** Characteristics of the relative change of rib waveguide's homogeneous sensitivity resulting from variations of opto-geometrical parameters: $H$, $w$, and $t$ (**a**) and $n_f$ (**b**). The description of the rib waveguide's parameters is given in the caption of Figure 1.

**Table 5.** Opto-geometrical parameters and homogeneous sensitivity of rib waveguides selected for assessment of the $\delta S_H$.

|  | $H$ [nm] | $t$ [nm] | $w$ [µm] | $S_H$ |
|---|---|---|---|---|
| $HE_{00}$ | 120.0 | 100.0 | 1.0 | 0.1194 |
| $EH_{00}$ | 175.0 | 100.0 | 1.0 | 0.1543 |

The question remains about the uncertainty of homogeneous sensitivity values calculated using a pair of $n_{eff}(t)$ characteristics. The following treatment was applied because the uncertainty of effective refractive indices is unknown. Let us consider two characteristic $n_{eff}(t)$, one for cover refractive index $n_c$ and the second for $n_c + \Delta n_c$. By increasing $t$ tiny, the values of $\Delta n_{eff}$ may be the same, even though both characteristics $n_{eff}(t)$, are monotonously decreasing, and should also give varying characteristics of $\Delta n_{eff}$. However, there are variances between the two levels due to limitations to the accuracy of the $n_{eff}$ values of $S_H$. This can be observed in the characteristics presented in Figure 14. The uncertainty of $S_H$ read from these characteristics is $\Delta S_H = 1.2 \cdot 10^{-4}$.

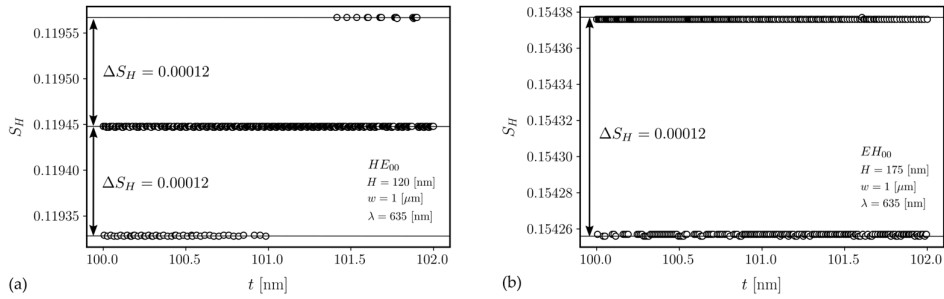

**Figure 14.** Homogeneous sensitivity of the rib waveguide's modes $HE_{00}$ (**a**) and $EH_{00}$ (**b**) in the limited variability range of $t$. Parameters $H$ and $w$ are given in Table 5.

## 4. Discussion

As has already been mentioned, the nature of variation of slab waveguide's homogenous sensitivity $S_H$ with the thickness of the waveguide film is well known. Each polarization has one value of the waveguide film thickness maximizing $S_H$. The comparison of maximum achievable homogeneous sensitivities for slab waveguides based on SOI, $Si_3N_4/SiO_2$, and $SiO_2$-$TiO_2$/BK7 reveals that SOI waveguides are the most sensitive to variations of $n_c$. This results from the fact that the refractive index of silicon $n_f = 3.4764$ is the highest compared to silicon nitride $n_f = 1.9963$ and $SiO_2$-$TiO_2$ $n_f = 1.7595$. However, considering the $HE_0$ mode, one can observe in Figure 4b and read from Table 2 that its maximum

sensitivity is reached for a waveguide film with a thickness of only $d_f$ = 36 nm. Typically, SOI wafers come with epitaxial Si layers of thickness ranging from 200 to 300 nm. The homogeneous sensitivity of the $HE_0$ mode in this range is lower than 0.05.

Considering the variation of rib waveguides $S_H$ with the parent slab thickness, it generally has the same character as the slab waveguides. Modes $EH_{00}$ have a single local maximum, as shown in Figure 10b. In most cases, the subsequent characteristics sharing the same $H$ are well separated. The separation is smaller only for those corresponding with parent slab thickness close to $d_{max}$ = 165 nm ($H$ = 160 and 175 nm). In the case of the $HE_{00}$ mode, which does not acquire the maximum for some intermediate rib height $t \in (0, H]$, the best thing is to look at the dependence on $H$ when $t = H$. Considering the variation of $S_H$ with $t$, it can be noted that for rib waveguides whose parent slab thickness is smaller than the thickness of the optimized slab waveguide ($H < d_{max}$), an increase in $t$ results in a decrease in $S_H$. The difference between the behavior of $HE_{00}$ and $EH_{00}$ modes shows up when the variation of $S_H$ with $t$ is considered for $H > d_{max}$. The sensitivity of $HE_{00}$ modes is the maximum for strip waveguides, whereas, in the case of $EH_{00}$ modes, the maximum is obtained for $t \in (0, H]$.

Comparing the characteristics presented in Figures 6 and 10a, we conclude that the character of the relation between the rib's width and the homogenous sensitivity of the strip waveguide depends on the relation between $H$ and $d_{max}$. The $S_H$ increases with $w$ if $H < d_{max}$ and decreases if $H > d_{max}$. This relationship is stronger for the $HE_{00}$ mode.

Homogeneous sensitivity $S_H$ is a derivative of the effective refractive index $n_{eff}$ concerning the refractive index of the cover $n_c$ (1). Assuming the increase of $n_c$ is small and constant, homogeneous sensitivity increases with the rise in $n_{eff}$. The magnitude of $n_{eff}$ increases with an increasing share of the optical power in the waveguide film, which has the highest refractive index. For this reason, we proposed building a relationship between the weighted filling factor $\Delta\Gamma_F^w$ given by (8) and $S_H$. As one can observe, the characteristics $\Delta\Gamma_F^w(S_H)$, presented in Figures 7 and 11, agree with the assumption that $S_H$ is the maximum if a weighted difference of the filling factor is the maximum. As a result of a small increase in $\Delta n_c$, for both polarizations, the optical power flows from the substrate to the waveguide film and cover ($\Delta\Gamma_F^s < 0$). The magnitude of homogenous sensitivity depends on how much optical power is drawn out from the substrate and how it is further divided between the waveguide film and the cover.

The characteristics of homogenous sensitivity relative change, $\delta S_H$, resulting from variations of rib waveguide's opto-geometrical parameters, reveal that $\delta S_H$ is greater than the uncertainty of $S_H$ for $n_f$. The rib waveguide's geometrical parameters shift the results in more minor $S_H$ changes. From the point of view of the mentioned material platforms, it poses a challenge for those platforms using multi-compound waveguide films, such as $SiO_2$-$TiO_2$. That is because the refractive index of such films depends on the environmental conditions prevailing during the fabrication process, e.g., humidity.

## 5. Conclusions

A rib waveguide's fundamental modes of polarizations, $HE$ and $EH$, can have higher homogenous sensitivity than the optimized slab waveguide. The values of the maximum increase over relevant sensitivity values for the optimized slab are minimal. And so, in the case of $HE_0$ and $HE_{00}$ modes, the homogeneous sensitivities of the optimized parent slab and rib waveguide are 0.119 and 0.120, respectively. In the case of the $EH_0$ and $HE_{00}$ modes, it is 0.154 and 0.155, respectively. The important difference between the magnitude of the response of the $HE_{00}$ and $EH_{00}$ mode to a variation of the cover's refractive index is that the $HE_{00}$ mode is most sensitive if the rib waveguide becomes a strip waveguide. Suppose the magnitude of $S_H$ is essential for the design of some integrated optics structure, which, for other reasons, should be based on strip waveguides. In that case, the $HE_{00}$ mode may have higher sensitivity. Structures comprising strongly bent waveguides, e.g., ring resonators,

may serve as an example. This is because the presence of a sacrificial layer results in high bending losses.

**Author Contributions:** Conceptualization, C.T.; methodology, C.T.; software, C.T. and P.K.; validation, C.T. and P.K.; formal analysis, C.T. and P.K.; writing—original draft preparation, C.T.; writing—review and editing, C.T. and P.K. All authors have read and agreed to the published version of the manuscript.

**Funding:** The research was co-financed by the Foundation for Polish Science from the European Regional Development Fund within the project POIR.04.04.00-00-14D6/18 "Hybrid sensor platforms for integrated photonic systems based on ceramic and polymer materials (HYPHa)" (TEAM-NET program).

**Institutional Review Board Statement:** Not applicable.

**Informed Consent Statement:** Not applicable.

**Data Availability Statement:** Not applicable.

**Conflicts of Interest:** The authors declare no conflicts of interest.

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
