# Peer review of "Evanescent Wave Sensitivity of Silica-Titania Rib Waveguides in the Single-Mode Propagation Regime"

_photonics, doi:10.3390/photonics10091065_

Round 1

Reviewer 1 Report

I have the following suggestions to improve the quality of the paper so that it can be acceptable after the revision. 

1) There is a continuous mixing of EH, HE, TM, and TE terms. Try to use one kind of abbreviations either (HE, EH) or (TE, TM). 

2) The author should emphasize on the importance of the waveguides designed on silica-titania platform. Why this platform is so attractive and also provide some recent literature on the devices designed on such platform and their performance. The literature is scarce, however, I found few recent papers on this topic such as: https://doi.org/10.3390/photonics10090978; https://doi.org/10.3390/photonics10020208. So that it can be verified that the sensitivity analysis presented in this paper is useful for some practical devices based on this platform. 

3) Are these results verified with numerical methods such as FEM, FDTD methods. What is the percentage of error? 

4) The abstract section should be modified and limit it to most important findings in the paper such as single-mode regions and the maximum sensitivity offered by the waveguide designs.

5) The author should provide a comparison of sensitivity obtained in this platform with other standard platforms so that it can be accessed that the sensitivity obtained in this work has some meaning. Because at this time, it is just a number which does not signify if its good value or bad. Also in conclusion section, the maximum sensitivity obtained for specific dimensions should be documented. 

None

Author Response

I am grateful to the Reviewer for the careful and thorough reading of the manuscript and constructive comments that helped me improve it.

In this file the remarks, rewritten in italics, are addressed in detail. In the revised manuscript, I highlighted corrected text parts with green background color. If, as a result of answering the given remark, the new input has been added to the revised manuscript, then in the answer, there is information on which lines should be sought for this new input.

Sincerely

Cuma Tyszkiewicz

Reviewer 2 Report

Reviewers Comments –

“Evanescent wave sensitivity of silica-titania rib waveguides in the single-mode propagation regime”

            The authors present a rigorous numerical approach in order to investigate the sensitivity improvements of a rib waveguide design vs a slab mode for evanescent refractive index sensing. The authors provide clear and lengthy explanations on the methodology and details explaining the design and the variations on waveguide geometry explored, which covers a large design space. Performed for the silica-titania material platform, the authors could provide better justification or details on the motivation for it as a material platform in this application space as it is not made directly clear in the introduction. The manuscript is well organized, the quality of writing is high, and the conclusions are clear. It is the opinion of the reviewer that the manuscript is of interest for the photonics community as it provides meaningful work that can be carried further towards the development of on chip bio sensors and can aid in the design of future fabricated components. Below is a brief list of comments which the authors may consider to strengthen the motivation for the manuscript with their line locations in red.

51 – Is this a commonly known technique used? If so a reference could be added possibly. It may not be so intuitive to readers that the sensitivities of the slab mode may be used to assess the influence of sensitivity metric from patterning the rib waveguide.

68 – The authors are strongly recommended to motivate their material choice as many platforms for bio sensing are tending towards more available platforms such as Si3N4 with VLSI fabrication. Since it was mentioned in the intro, the material needs to be slightly introduced with reasoning for the choice of platform. Although based on previous work and referenced, a brief description of the rationale is necessary for the paper. For example, BK7 wafers have more inherent curvature and less scalability than Si substrates. This contrast should be discussed. Also, in general the claims for sensitivity are written to be in a general sense. The authors should speculate what would change about their analysis when considering other material systems and if their results would still hold true (for example is a TM rib waveguide always more sensitive in Silicon nitride than a slab mode). Further to this point, it could be possible a reader considers why it is even valuable to prove that an etched waveguide is better than a slab mode for sensing using this material platform. This needs to be clear. 

225 – The case where t=H can be referred to as fully etched, or a strip waveguide if I am not mistaken. This may be useful language to use in the paper for the reader to make easier physical connections with the parameter space. I see it written sometimes, but not every time.

246 – The axes for Figure 6 a difficult to read. if formatting is possible, it is recommended to increase the size of the font. Additionally, the placement of (a) and (b) in the figure is not so intuitive

Author Response

(The authors gave the same response as above.)

Reviewer 3 Report

The paper entitled ‘Evanescent wave sensitivity of silica-titania rib waveguides in 2 the single-mode propagation regime’ theoretically studies the influence from morphology of rib waveguide to the homogeneous sensitivity when using the waveguide as a sensor. They provide solid numerical analysis from their study on various modes supported by the waveguide. They have ample results to support their claims, but their writing needs to be improved. There are also some minor comments I would like to provide. Therefore, I recommend a minor revision.

Here is the list of my major comments:

1.     The authors should provide more justifications on why the investigation on rib waveguide is significant at the beginning of the introduction. How does it compare with other types of waveguides speaking of refractive sensing?

2.     The author should provide more information on why they choose silica-titania as the waveguide material.

3.     Given the material, the author should comment on how easy is the refractive index of the material, and how accurate the fabrication can be achieved.

4.     Also, the author should comment on how does the fabrication error will affect their conclusion, since they are studying the structures with sub-10 nm size variations and high refractive index sensitivity.

5.     The author should also provide some potential applications using the design they have and make comparison to other published works.

Minor comments:

There are several long sentences in the manuscript which is hard to follow. Reading those sentences can be arduous. They author might consider to break them into short and concise sentences. For example, the very first sentence in the abstract is confusing.

There are several long sentences in the manuscript which is hard to follow. Reading those sentences can be arduous. They author might consider to break them into short and concise sentences. For example, the very first sentence in the abstract is confusing.

There are also grammar errors in the manuscript. The author may consider double check the manuscript.

Author Response

(The authors gave the same response as above.)

Round 2

Reviewer 1 Report

I am willing to accept the paper in its current form. 

none.